# Flat out Fabulous: How Barbie's foot posture and occupations have changed over the decades, and the lessons we can learn

Cylie Williams[1,2]*, Kristin Graham[2], Ian Griffiths[3], Suzanne Wakefield[4], Helen Banwell[2]

1 School of Primary and Allied Health, Monash University, Frankston, Victoria, Australia, 2 Allied Health and Human Performance, University of South Australia, Adelaide, South Australia, Australia, 3 Centre for Sports & Exercise Medicine, Queen Mary University of London, London, United Kingdom, 4 Barbie Collector, Melbourne, Victoria, Australia

* Cylie.williams@monash.edu

## Abstract

### Objectives

To explore the correlations/relationships between foot posture, equity and diversity, employment, and time in Barbie Land.

### Design

A retrospective audit of the Barbie population (or their data from online catalogues) using a customised FEET system. That is **F**oot posture (flat or equinus); **E**quity (diversity and inclusion (EDI)); **E**mployment (fashion vs employed); and **T**ime period (decade of manufacture).

### Setting

Barbie Land (Online Barbie catalogues of doll types).

### Participants

2750 Barbies and friends who lived in Barbie Land between 1959 to June 2024.

### Main outcome measures

Over time there was a decreased prevalence in equinus foot posture from 100% in the first time period to 40% in the last. Barbie's flat foot posture had a very strong positive correlation with employment (r = 0.89, 95% Confidence Interval (95%CI) = 0.50 to 1.29), and time point (r = 0.85, 95%CI = 0.40 to 1.31), while equinus foot posture had a very strong positive correlation with fashion (r = 0.99, 95%CI = 0.87 to 1.11). Similarly, equity (diversity) had a very strong positive correlation with fashion (r = 0.98, 95%CI = 0.82 to 1.15), and strong positive correlation with employment (r = 0.79, 95%CI = 0.26 to 1.33).

**Data availability statement:** All relevant data are within the manuscript and its Supporting Information files.

**Funding:** The author(s) received no specific funding for this work.

**Competing interests:** No authors have competing interests.

## Conclusion

Barbie's equinus foot posture is directly related to her high heel wearing. Barbie models her footwear choice based on task demands, being flat footed and wearing flat shoes when she needs to work on her feet, be physically active or more stable. Given Barbie is known to reflect societal norms, we contend this is most likely true for most high-heel wearers. While Barbie has moved with the times, it appears footwear health messaging about high heel wearing needs to catch up. Health professionals castigating high heels through public messaging, should remember that emphasising health benefits consistently drives positive behaviour change, over highlighting negative consequences. Barbie clearly makes sensible determinations regarding her body autonomy; high heel wearers should have that same ability.

## Introduction

The success of the 2023 Barbie movie (Universal Studios, Warner Bros Pictures) reinvigorated analysis on Barbie's standing within social and cultural contexts. With a premise of "thanks to Barbie, all problems of feminism and equal rights have been solved" [1], 'Barbieland' is a seemingly perfect matriarchy. This is at least until "Stereotypical Barbie" (played by Margo Robbie) experiences an existential crisis presenting as flat feet. The equinus foot posture (where the ankle lacks adequate dorsiflexion or tip toe position) is iconically associated with Barbie. So much so, that standing with feet flat on the floor was not only 'gag-inducing' in the movie, it launched the #Barbiefeetchallenge. This challenge asked participants to mimic the fixed equinus foot posture when exiting high-heeled shoes. While this seemingly celebrates the foot posture itself, it is a clear statement on Barbie's preferred choice of high heeled footwear.

Outside of Barbie Land, aka 'the real world', health professionals have discouraged the use of high heeled footwear since the early 1900's [2,3]. Health professionals often link high heel footwear as the cause of bunions [4,5], knee osteoarthritis [6], plantar fasciitis [7,8] and low back pain [9]. Yet many of these health conditions are highly prevalent in the general population with low or no high heel use [10–12]. It is also purported that excessive high heel wear results in ankle equinus, a foot deformity, such as Barbie commonly displays. While there is no disputing that ankle equinus is damaging to foot health and function, there is no known direct link between the use of high heeled shoes and ankle equinus.

The majority of studies investigating high-heeled shoes only investigate the immediate and short-term impact [13], and often in non-high heel wearers [14]. Or studies are focused where shoes with a heel pitch are atypically worn such as during high-intensity or sports related tasks [15,16]. From these investigations, we understand high heel footwear has immediate and significant effects on walking gait; including reduced speed, shorter steps, and increased lower limb instability [17]. However, a recent meta-analysis exploring high heeled shoe wearing and lower extremity biomechanics underlined the low levels of data available and the weak

nature of measures of wearing impact. The cited review called for longitudinal research covering a wider range of heel heights to better understand their impact [18]. However, given there appears to be no known long-term studies on the use of high heeled shoes that mimics casual and considered use by participants, it begs the question if this evidence is relevant to the bulk of the population or is the choice Barbie makes with footwear, reflective of real life.

Since Mattel™ launched Barbie in 1959, they have sold over 1 billion dolls worldwide [19] with 92% of American girls aged 3–12 years having owned a Barbie [20]. Her global sustained iconic status is a product of her ability to be responsive to changing societal norms, aspirations, and identities [21,22]. Barbie has demonstrated her commitment to increasing inclusivity and diversity through expanding the racial and ethnic diversity, and disability representation of herself and friends. Barbie is also committed to female empowerment, demonstrating "Girls can do anything" through undertaking many traditionally male-dominated careers. In 1965, before man landed on the moon, Barbie became a Special Collector's edition astronaut, and in 1973 when 91% of doctors in the US were male, Barbie became a surgeon [23].

In addition to Barbie's roles, her body morphology has changed over time, partially as a result of societal expectations [22], sparking many philosophical debates [24–26]. However, her ankle equinus has not been explored in detail. Given the interest in "flat feet" generated by the Barbie movie, our hypothesis was that if Barbie has truly increased her representation of societal roles such as employment, she would make footwear choices (and by extension, related foot postures) based on what best enables her to undertake these roles. We were also interested in Barbie's diversity, both in ethnicity and ability, and if there was any impact on foot posture.

## Materials and methods

This research was a retrospective audit from publicly published data aka Barbie and friends living in Barbie Land. We used published Barbie collector catalogues [27,28], sales sites (e.g. eBay) and personal Barbie collections as reference materials to extract data. We used multiple catalogues as a method to cross check the accuracy and completeness of Barbie's names across the years. We used the images in the sales sites and personal collections to cross check accuracy of foot postures and develop images as examples for the FEET system. During development of our study protocol, the authors met online to discuss their personal reflexivity and its influence on this research and findings. There were four authors who owned and played with at least one Barbie during childhood, one author who wears high heel footwear at least three days per week in the workplace, three who have intermittently worn high heels in the past, and one who has have never worn a heel higher than 12mm.

We limited audit data inclusion to Barbie and her similar aged friends (e.g., Christie, Francie), regardless of if their ankles were fixed or moveable. To ensure outcomes related to the 'mass market, we excluded any Barbie or friend that had limited international reach, such of those marked as special/limited/collector editions or aligning with Television series (e.g., Dolls of the World, Crayola, Sponge Bob Square Pants), those modelled on popular actual people (e.g., Audrey Hepburn), when Barbie was packaged together with Ken, Barbie's younger sisters and their friends (e.g., Skipper), any who were Mermaids without legs/feet or ballerinas with fully moulded feet into shoes, Barbies with soft bodies or those with horses or vehicles. We also excluded Barbie released in only a single country or only one retail outlet (e.g., USA only releases for Sears, Target, or Walmart). We made these decisions to ensure we only categorised the most accessible, representative and mass-produced Barbies available to people throughout the world. This ensured generalisability across international distribution points rather than Barbie dolls that were produced for a niche or single geographical market. This also ensured representation accuracy across the referenced online catalogues and multiple reference points to support the validity of the developed data collection systems. Many of the special/limited edition dolls, or those not widely distributed had limited online pictures which would have introduced potential misclassification of foot posture.

We categorised the foot posture as equinus (>10 degrees) or flat (≤10 degrees) (Fig 1). We also categorised other aspects of Barbie, using our author developed FEET system designed to categorise characteristics of interest, including

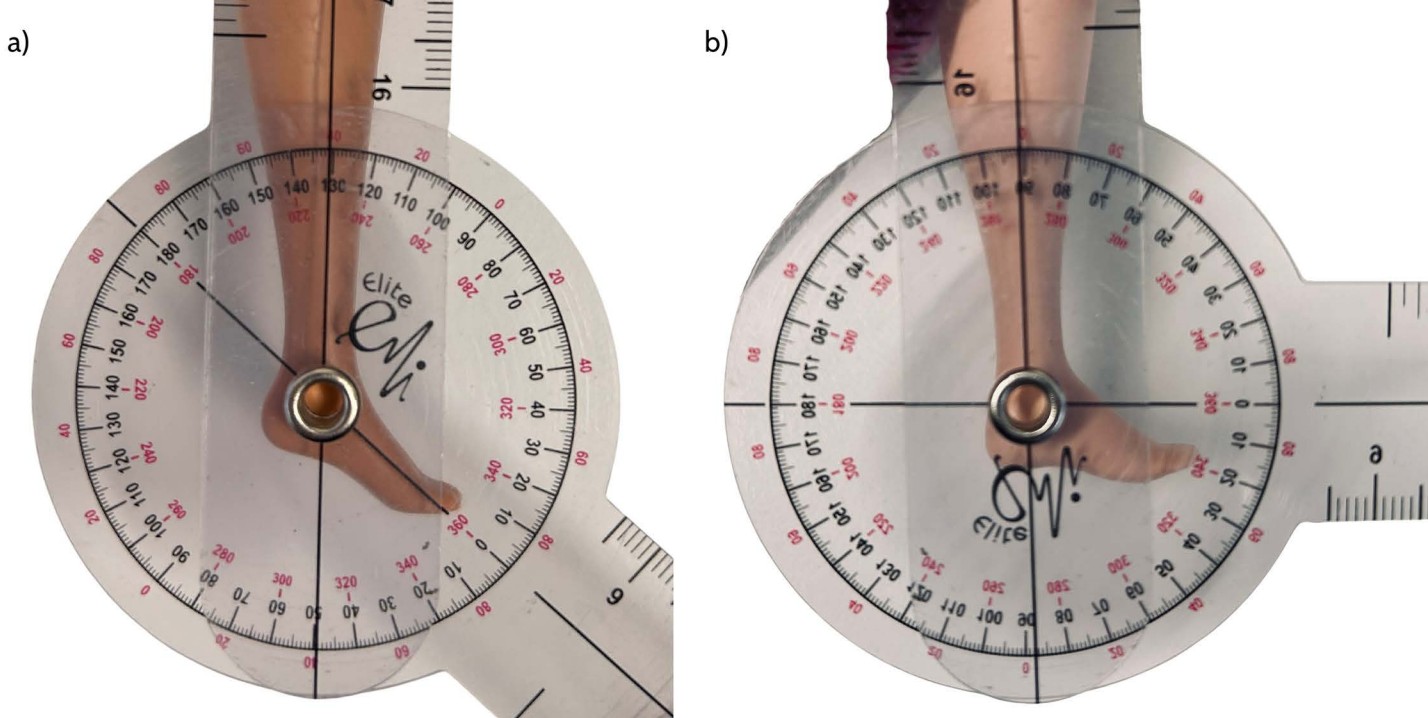

**Fig 1. Barbie's ankle equinus measured at 50 degrees as equnius posture (a) and flat foot posture measured at 0 degrees (b), with a handheld goniometer.**

*Foot posture* (flat or equinus); *Equity (diversity and inclusion* (EDI)); *Employment* (fashion vs employed) and, *Time period* (decade of manufacture) (Table 1). As data extracted was binary (e.g., yes, no), we did not formally test reliability of this purpose-built and simple rating system. However, during its development, all authors met to clarify and refine the definitions and terminology for clarity. Then all data were extracted by a single author (CMW) into the custom developed spreadsheet, and 10% (n = 275) of the data extracted were independently double checked by a second author (IG, KG or HB) to ensure correct categorisation against the FEET system.

We included consumer and community involvement within this research. We involved an independent Barbie expert to share her love of Barbie and Barbie collection for verification of Barbie inclusion categories and ensure correctness in alignment with the FEET system (Fig 2). We also confirmed both Barbie inclusions and FEET system with an adolescent female within one author's household to ensure our categorisation reached 'yeet' status with this representative group.

Data was entered into Microsoft® Excel for Mac (Version 16.87), then transferred to Stata 15.0 (Release 15, StataCorp, 2023). to calculate the summary statistics across each timepoint and data analysis. All data except for Foot posture (flat) and Equity (disability) were normally distributed. When exploring the kurtosis of the data, we found the Foot posture (flat) was slightly platykurtic and Equity (disability) to be leptokurtic. Spearman correlations were conducted to fully explore the impact of the limited data within the Equity (disability) and deviation to normality before deciding on a final model [28] Due to minimal differences between the Pearson and Spearman correlation coefficients and the Pearson correlations coefficients being more conservative, only the Pearson coefficients are presented [29]. Pearson correlation coefficients were calculated as a measure of the strength of the association where 0.00 to 0.19 was very weak, 0.20 to 0.39 was weak, 0.4 to 0.59 was moderate, 0.6 to 0.79 was strong and 0.80 to 1.00 was very strong correlation [30]. We also reported the 95% confidence intervals (95% CI). Julius AI ([www.julius.ai](http://www.julius.ai)) was used to develop a correlation heatmap graphic, and graph of

**Table 1. The FEET system of barbie categorisation.**

| Category | Definition | Category examples |
|---|---|---|
| **F**oot posture | *Flat* = a fixed ankle ankle of ≤ 10 degrees<br>*Equinus* = a fixed ankle angle of > 10 degrees | *Flat*:<br>Barbie Kid Doctor<br>Barbie Nurse<br>*Equinus*:<br>Barbie Pony tail<br>Barbie Hair Happenin's |
| **E**quity (diversity and Inclusion) (EDI) | Barbie visually represented a person of colour, or displayed disability inclusive of assistive technology use (e.g., wheelchair use). | *Diversity*:<br>Barbie representative as Hispanic, Black or Asian<br>*Disability*:<br>Barbie with Down Syndrome<br>Barbie with an above knee amputation |
| **E**mployment or fashion role | Employment required Barbie's role to be focused on employment or representation of an activity associated with fee for service or sponsorship.<br>Fashion included Barbie's role identified as changing appearance, leisure based or conducting fashion focused activities. | *Employment:*<br>Paediatrician Barbie<br>Eye doctor Barbie<br>Barbie Gold Medal Skier<br>*Fashion:*<br>Malibu Barbie<br>Twirly curls Barbie<br>Fashion jeans Barbie |
| **T**ime Period | 10-year period in which Barbie was released | 1960 (inclusive of 1959)<br>1970's<br>1980's<br>1990's<br>2000's<br>2010's<br>2020's (inclusive of June 2024) |

percentage of total dolls over time for each of the datasets from the FEET system. There were no missing data as our unit of measure was Barbie, so she was included even if she only had one foot.

## Results

We catalogued 2750 Barbie and her friends (S1 File). There was a decreased prevalence in Barbie's equinus foot posture from 100% (n = 48) in the first time period, to 40% in the last 4 years (n = 134 of 332) (Table 2, Fig 3).

Equity, diversity and disability was observed from Barbie's conception, with 5% of Barbies between 1959–1969 presented as "non-stereotypical" Barbie, with representation expanding to 46% in recent years. Barbie's use of assistive technology also increased from a single Barbie wearing an above knee prothesis in the time period between 2010–2019, to multiple wheelchair users (n = 3), along with another with an above knee amputation, and one who uses ankle foot orthoses in 2020–2024 (Table 2, Fig 3).

Flat foot posture had a very strong positive correlation with employment (r = 0.89, 95% CI = 0.50 to 1.29), and time point (r = 0.85, 95% CI = 0.40 to 1.31), while equinus foot posture had a very strong positive correlation with fashion (r = 0.99, 95% CI = 0.87 to 1.11). Similarly, equity (diversity) had a very strong positive correlation with fashion (r = 0.98, 95% CI = 0.82 to 1.15), and strong positive correlation with employment (r = 0.79, 95% CI = 0.26 to 1.33). There was moderate positive correlation of flat foot posture with disability (r = 0.67, 95% CI = 0.02 to 1.32) and diversity (r = 0.58, 95% CI = -0.14 to 1.29), however the distribution of the 95% CIs of these results mean any relationships are less consistent (Fig 4).

## Discussion

This is the first-known exploration of the novel FEET system of Barbie, aiming to explore the relationship between Barbie's roles, representations and her foot morphology. The strongest correlations with a flat foot posture in our observations

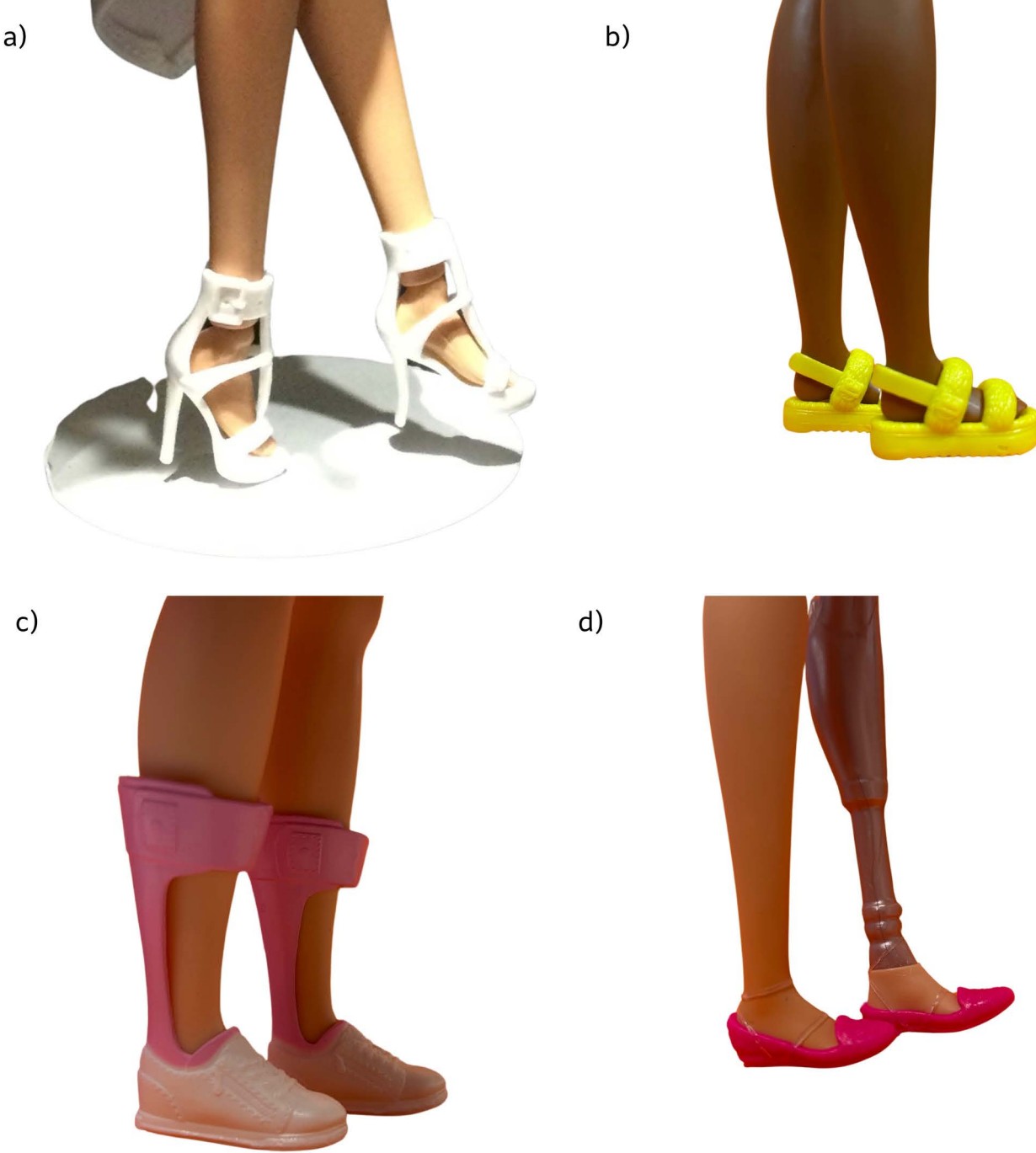

**Fig 2. Foot posture examples from personal collections – a) Barbie Made to Move (equinus), b) Barbie Fashionista #210 (flat), c) Fashionistas Barbie heart DownSyndrome (Assistive technology and flat) d) Interior designer Barbie with above knee amputation (Assistive technology and flat).**

**Table 2. Frequency outcomes of an audit of dolls manufactured between 1959–2024 aligned with The FEET system of Barbie categorisation.**

| Timepoint | Barbie and friends N | Foot posture | | Equity | | Employment | |
|---|---|---|---|---|---|---|---|
| | | Flat n (%) | Equinus n (%) | Diversity n (%) | Inclusion relating to disability requiring Assistive Technology use n (%) | Fashion n (%) | Employed n (%) |
| 1959–1969 | 44 | 0 (0%) | 44 (100%) | 2 (5%) | 0 (0%) | 44 (100%) | 0 (0%) |
| 1970–1979 | 94 | 0 (0%) | 94 (100%) | 13 (14%) | 0 (0%) | 89 (95%) | 5 (5%) |
| 1980–1980 | 153 | 0 (0%) | 153(100%) | 78 (51%) | 0 (0%) | 125 (82%) | 28 (18%) |
| 1990–1999 | 367 | 34 (9%) | 333 (91%) | 185 (50%) | 0 (0%) | 298 (81%) | 69 (19%) |
| 2000–2009 | 879 | 56 (6%) | 823 (94%) | 384 (44%) | 0 (0%) | 771 (88%) | 108 (12%) |
| 2010–2019 | 883 | 251 (28%) | 632 (72%) | 339 (38%) | 1 (<1%) | 603 (68%) | 280 (32%) |
| 2020–2024 (mid-year) | 330 | 198 (60%) | 132 (40%) | 152 (46%) | 5 (2%) | 222 (67%) | 108 (33%) |

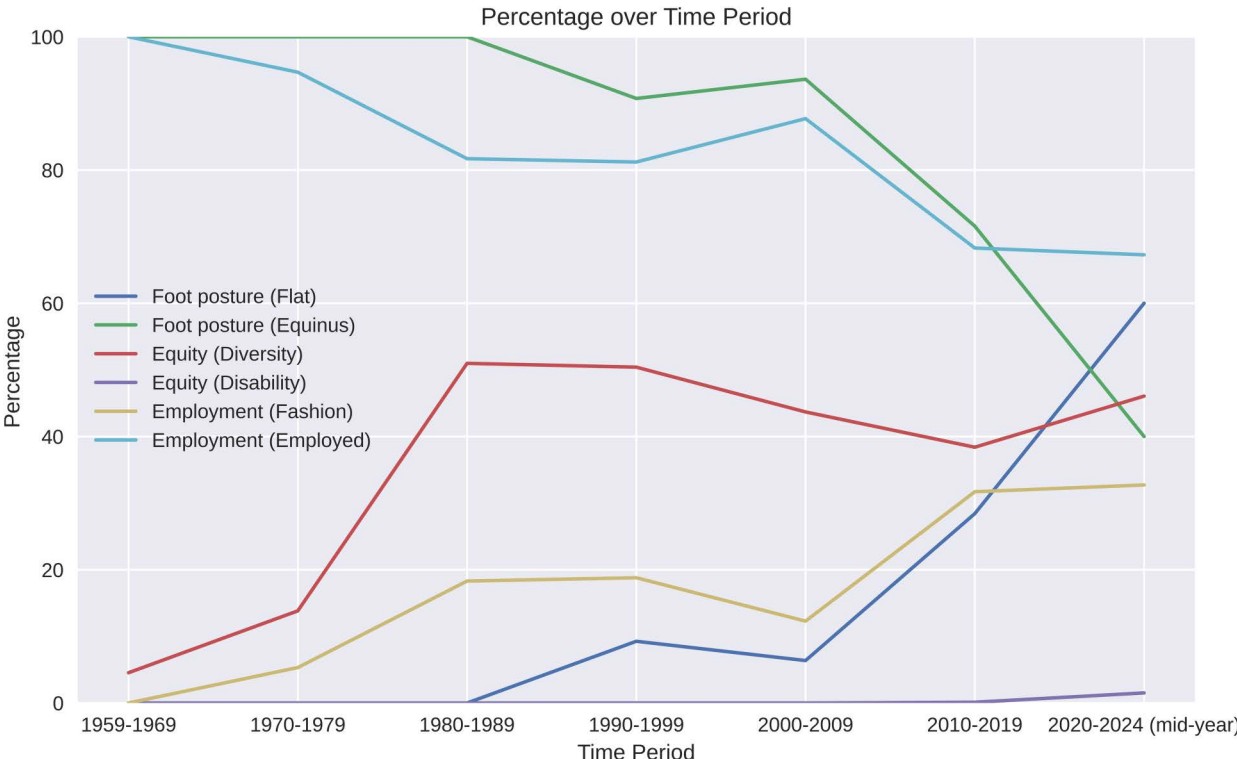

**Fig 3. Percentages of Barbie in each of the FEET system categories over time.**

was with employment, closely followed by time periods. While the increase in Barbie's use of assistive technology was minimal, Barbie in her wheelchair was still rocking killer heels with equinus foot posture, while those with an above knee amputation had a flat foot posture.

The change in workforce roles of Barbie mimics the change in women's employment across the world [31,32]. Aligning with these changes is the increase in employment equity enshrined in discrimination laws, particularly in the USA

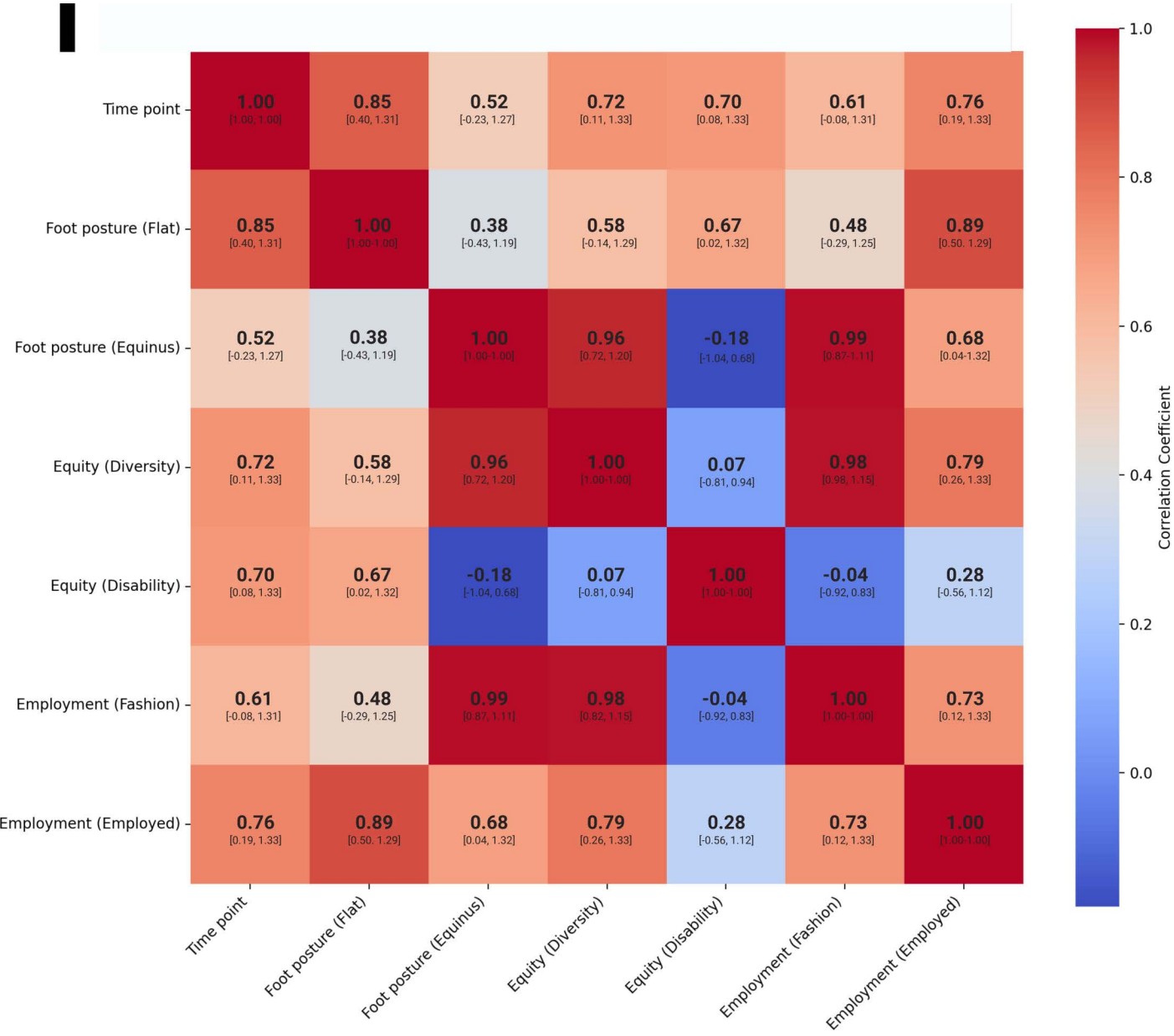

**Fig 4. Heat map of correlations of the FEET system with 95% CI values within brackets.**

where Barbie originates. The frequency of employment roles rapidly increased from the 1980's, potentially aligning with a landmark change in USA laws where women could independently own business without a male relatives permission (H.R.5050 - Women's Business Ownership Act of 1988) [33]. Our observed strong relationship between flat foot posture and time, suggests these changing societal roles and laws were reflected in Barbie's foot posture.

The very strong correlation between flat foot posture and employment sits well with evidence observing Barbie 'evolving' away from prioritising fashion over her career [34]. Just as her facial features and body shape have changed over time, her expanding workforce roles create a requirement for prolonged standing, increased walking speed, and greater postural stability. As a role model to many young people, it is encouraging to see Barbie making foot posture and footwear

choices that best enable her participation in employment and physical activity. Given a third of the population is not getting enough physical activity [35], footwear change might just be one of many opportunities to promote increased movement. This may be a more appropriate approach to guiding footwear advice, than the castigation of high heel wearing often seen in medical and journalistic websites [36,37].

We found Barbie's foot posture change over time unexpected and fascinating, and perhaps something with deeper meaning or a Mattel executive direction we are unaware of. It is unclear why high heels are inherently linked with women's fashion, given they originated in the male workforce and society [38]. However, there are clear changes in wearing patterns over time and preferenced wearing of high heel footwear linked with psychosexual benefits [39]. In the movie, Barbie's friends found her flat feet nauseating, even Barbie made numerous references to her altered movement ability with flat feet, and not always in a positive light. Too often the focus on footwear health messaging regarding high heel use is negative, however, as Ruth Handler (Barbie's inventor) reminds Barbie in the movie, being human is uncomfortable. The use of high heel shoes in fashion is a long held custom, and a fashion choice that many wearers enjoy and prefer. Critically, any potential adverse impacts from wearing high heels are constrained to those who make this choice, and the frequency of this choice.

This research has its limitations. It is quite possible by limiting the Barbie collections to exclude special collections, the prevalence of Barbies with flat feet have been increased. Similarly, when comparing the correlation models, the use of Spearman correlation coefficient increased the strength of some relationships. Given the inclusion criteria, and minimal difference, we reported the more conservative of the two models. Also, while these correlations are strong and suggest important relationships, they should be interpreted cautiously. The sample size, even with variation across the time period is robust, allowing for stable estimates [40]. Correlation does not imply causation, and other factors not captured in this dataset could be influencing the relationships we observed. Future research is required to fully explore the drivers of Barbie's foot morphology choices and whether Barbie is a reflection of, or an influencer of female roles and fashion.

## Conclusions

We found a very strong correlation of flat foot posture with time and employment. This suggests Barbie Land has a dynamic environment with evolving employment patterns and social policies. Maybe as Barbie Land has evolved to having far more dolls with flat feet postures, footwear messaging should also evolve. That is, laying off on the potential doom and gloom public health messaging of wearing high heels and what they might (but might not) do to the body. Instead, trusting that most high-heel wearers, predominantly women, will pick footwear based on task demands. Health professionals castigating high heels through public messaging, should remember that emphasising health benefits consistently drives positive behaviour change, over of highlighting negative consequences. Barbie clearly makes sensible determinations regarding her body autonomy; high heel wearers should have that same ability.

## Supporting information

**S1 File.  Barbie audit data.**
(XLSX)

## Author contrivbutions

**Conceptualization:** Cylie Williams, Ian Griffiths, Suzanne Wakefield, Helen Banwell.

**Data curation:** Cylie Williams, Helen Banwell.

**Formal analysis:** Cylie Williams, Helen Banwell.

**Investigation:** Kristin Graham.

**Methodology:** Cylie Williams, Kristin Graham, Ian Griffiths, Suzanne Wakefield, Helen Banwell.

Software: Cylie Williams.

Validation: Cylie Williams, Kristin Graham, Ian Griffiths, Suzanne Wakefield, Helen Banwell.

Visualization: Cylie Williams.

Writing – original draft: Cylie Williams, Ian Griffiths, Suzanne Wakefield, Helen Banwell.

Writing – review & editing: Cylie Williams, Kristin Graham, Ian Griffiths, Suzanne Wakefield, Helen Banwell.

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
