## [Decision Letter · Decision Letter 0]

5 Feb 2025

PONE-D-24-58543Flat out Fabulous: How Barbie’s foot posture and occupations have changed over the decades and the lessons we can learn.PLOS ONE

Dear Dr. Williams,

Thank you for submitting your manuscript to PLOS ONE. After careful consideration, we feel that it has merit but does not fully meet PLOS ONE’s publication criteria as it currently stands. Therefore, we invite you to submit a revised version of the manuscript that addresses the points raised during the review process.

We look forward to receiving your revised manuscript.

Kind regards,

Yaodong Gu

Academic Editor

PLOS ONE

Journal Requirements:

Reviewers' comments:

Reviewer's Responses to Questions

**Comments to the Author**

1. Is the manuscript technically sound, and do the data support the conclusions?

Reviewer #1: Yes

Reviewer #2: No

2. Has the statistical analysis been performed appropriately and rigorously? 

Reviewer #1: Yes

Reviewer #2: No

3. Have the authors made all data underlying the findings in their manuscript fully available?

Reviewer #1: Yes

Reviewer #2: No

4. Is the manuscript presented in an intelligible fashion and written in standard English?

Reviewer #1: Yes

Reviewer #2: No

5. Review Comments to the Author

Reviewer #1: Reviewer comments:

1. The introduction emphasizes Barbie's status as a social symbol, but does it need further explanation as to why "foot posture" was chosen as the core entry point of the study, rather than other physical or professional characteristics?

2. When introducing the impact of high-heeled shoes on health and the importance of changes in foot posture, the current research on the risk of lower limb injury caused by high-heeled shoes changing ankle joint function is unclear. "New insights optimize landing strategies to reduce the lower limb injury risk" (https://doi.org/10.34133/cbsystems.0126). The study highlights that ankle function is critical for lower limb stability and reducing the risk of injury during exercise. Studies have shown that changes in foot posture not only affect gait stability, but also reduce potential injury risk by optimizing impact force absorption. These findings provide a theoretical basis for understanding how different footwear choices adapt to functional requirements. Combined with the change in Barbie's foot posture from horseshoe to flat foot, this study provides scientific support for its biomechanical significance and further highlights the correlation between task demands and foot functional adjustment.

3. The authors used a custom FEET system to assess Barbie's foot posture and variety. Is the system proven? Can you provide evidence on the reliability and consistency of the classification criteria?

4. Limited edition or special versions of Barbie (e.g. TV characters, limited collections) were excluded from this study. Does this bias the generality of the findings? What explains the scientific basis for these exclusion criteria?

5. Data mainly comes from online catalogs and personal collections. Is it possible that there are systematic biases in the accuracy and completeness of this information? Is the impact of differences in these sources on the results considered?

6. The results show that Barbie's flat foot posture is highly correlated with occupation, but these correlations are based on statistical analysis only. Can more care be taken to avoid interpreting statistical correlations directly as causation?

7. Although the charts (such as Figure 3 and Figure 4) show time and correlation trends, can more specific data annotations (such as standard errors or confidence intervals) be provided in the charts to support the confidence of the results?

8. Barbie's sample size varies widely across time periods (e.g., there were only 330 Barbie samples in 2020-2024 compared to 879 in 2000-2009). Could this uneven distribution of samples affect the results of statistical analysis?

9. The study notes that "changes in Barbie's foot posture may reflect dynamic changes in social employment patterns and policies." Can this inference be further explored in a broader sociological or culturological context?

10. The discussion section suggests that "health professionals should reduce negative perceptions of high heels." Is there a lack of direct evidence to support this suggestion? Does more literature or further analysis of Barbie data need to be done to support this?

Reviewer #2: I have read through this paper several times and spent a fair amount of time reflecting on the meaning of this paper as well as the potential contribution it can make to the literature. In my opinion, the work has no scientific value and falls within the realm of fantasy and speculation. It adds nothing from a clinical point of view. Therefore, I do not recommend the work for further proceedings.

6. PLOS authors have the option to publish the peer review history of their article (what does this mean? ). If published, this will include your full peer review and any attached files.

**Do you want your identity to be public for this peer review?** For information about this choice, including consent withdrawal, please see our Privacy Policy .

Reviewer #1: No

Reviewer #2: No

---

## [Author Response · Author response to Decision Letter 1]

3 Mar 2025

Please see detailed tracked changed uploaded with the manuscript.

---

## [Decision Letter · Decision Letter 1]

26 Mar 2025

PONE-D-24-58543R1Flat out Fabulous: How Barbie’s foot posture and occupations have changed over the decades and the lessons we can learn.PLOS ONE

Dear Dr. Williams,

Thank you for submitting your manuscript to PLOS ONE. After careful consideration, we feel that it has merit but does not fully meet PLOS ONE’s publication criteria as it currently stands. Therefore, we invite you to submit a revised version of the manuscript that addresses the points raised during the review process.

We look forward to receiving your revised manuscript.

Kind regards,

Yaodong Gu

Academic Editor

PLOS ONE

Reviewers' comments:

Reviewer's Responses to Questions

**Comments to the Author**

1. If the authors have adequately addressed your comments raised in a previous round of review and you feel that this manuscript is now acceptable for publication, you may indicate that here to bypass the “Comments to the Author” section, enter your conflict of interest statement in the “Confidential to Editor” section, and submit your "Accept" recommendation.

Reviewer #1: (No Response)

Reviewer #3: (No Response)

2. Is the manuscript technically sound, and do the data support the conclusions?

Reviewer #1: Partly

Reviewer #3: Partly

3. Has the statistical analysis been performed appropriately and rigorously? 

Reviewer #1: Yes

Reviewer #3: I Don't Know

4. Have the authors made all data underlying the findings in their manuscript fully available?

Reviewer #1: Yes

Reviewer #3: No

5. Is the manuscript presented in an intelligible fashion and written in standard English?

Reviewer #1: Yes

Reviewer #3: Yes

6. Review Comments to the Author

Reviewer #1: Review comments:

1. At present, the introduction of this paper mentions the effects of high heels on gait and foot health, but there is no discussion on the correlation between foot posture and lower limb injury risk. This results in a weak biomechanical background, leaving the impact of Barbie's foot posture evolution lacking scientific evidence. Studies have shown that foot posture and landing strategy have an important impact on the risk of lower limb injury, for example (New insights optimize landing strategies to reduce the lower limb injury risk, https://doi.org/10.34133/cbsystems.0126). This study highlights that foot posture is not just a fashion issue but can have a profound impact on health. Therefore, it is suggested that the authors consider this study to explore the evolution of Barbie's foot posture, which can also provide a reference for understanding real-world footwear choices and lower limb health.

2. The sample selection criteria (Barbie's screening criteria) are subjective. for example, the author's reference to limiting the number of Barbie dolls "for our own sanity" falls short of academic rigor. Please provide more objective screening criteria and remove subjective statements. For example, clearly explain why certain Barbie Dolls were excluded from the study.

3. The statistical method part is not detailed enough, for example, the premise assumptions of correlation analysis (such as whether variables obey normal distribution) are not explained. It is suggested that in the statistical analysis part, the data processing process should be explained in detail, and the applicability and limitations of the analysis method should be discussed.

4. Discussion section Some of the content is repeated, such as emphasizing the relationship between Barbie's foot position and career choice many times without exploring the possible mechanism, please simplify the discussion section, avoid repetitive content, and focus on the uniqueness of the study results and its scientific contribution.

Reviewer #3: This study exhibits serious flaws in methodological design and data reliability. The data sources primarily relied on publicly available catalogs, sales websites, and private collections, lacking scientific rigor and systematicity. This approach introduces significant selection bias and incomplete information risks, with no means to verify data authenticity or accuracy. Foot posture assessment was based solely on subjective image classification without objective measurements, further undermining the study’s credibility. The proposed FEET system, while innovative, lacks reliability and validity testing. Its classification criteria are overly simplistic, categorizing foot posture merely as "horseshoe foot" and "flat foot" without accounting for intermediate states or potential misclassifications. Data exclusion criteria were inadequately justified, particularly the arbitrary removal of special and limited-edition Barbie dolls, which lacks scientific rationale and severely limits the external validity of the findings. Statistical analysis relied exclusively on Pearson’s correlation coefficient, with no normality testing or control for confounding variables, rendering the methodology overly simplistic. In summary, the study suffers from critical weaknesses in data sourcing, research methodology, analytical approach, and ethical compliance, casting strong doubt on the reliability and scientific merit of its conclusions.

7. PLOS authors have the option to publish the peer review history of their article (what does this mean? ). If published, this will include your full peer review and any attached files.

**Do you want your identity to be public for this peer review?** For information about this choice, including consent withdrawal, please see our Privacy Policy .

Reviewer #1: No

Reviewer #3: No

---

## [Author Response · Author response to Decision Letter 2]

2 Apr 2025

Please note – all pages and lines are referenced to the tracked version

Reviewer 1:

At present, the introduction of this paper mentions the effects of high heels on gait and foot health, but there is no discussion on the correlation between foot posture and lower limb injury risk. This results in a weak biomechanical background, leaving the impact of Barbie's foot posture evolution lacking scientific evidence. Studies have shown that foot posture and landing strategy have an important impact on the risk of lower limb injury, for example (New insights optimize landing strategies to reduce the lower limb injury risk, https://doi.org/10.34133/cbsystems.0126). This study highlights that foot posture is not just a fashion issue but can have a profound impact on health. Therefore, it is suggested that the authors consider this study to explore the evolution of Barbie's foot posture, which can also provide a reference for understanding real-world footwear choices and lower limb health.

Response:

We note that this was the same comment from the first review and wonder if it was a cut and paste error given its similarity. Our original response agreed with of the overarching statement and we supported this with additional of articles.

As per last revision, we counter when we reviewed the manuscript Reviewer 1 offers, it is a modelling article based on six healthy male and landing strategies from a height and the impact on ACL injury, an injury we are not aware of being linked with high heel footwear use as people usually wear for walking, not jumping or playing sport. This offered reference has no discussion on high heel footwear. This offered reference also proposes a biomechanical model for injury prediction when landing in a set position, therefore not aligning with the aim of this article being walking and working in a high heel shoe.

As we previously responded, a person landing from a height has different impact to that of everyday walking, even in high heels. This article also within its own limitations described unknown transferability of results to females, who are the predominant wearer or high heels. We have also provided additional references for the challenge Rev 1 offers – there is no link between high heel wearing and equinus, the most recent systematic review reporting more longitudinal data required if this myth is to continue.

During the first review we provided additional evidence relating to infrequent high heel wearing. Again, we agreed and clarifyed that our statement around the impact of high heeled shoes on the lower limb was originally unclear. But then we then provided referenced impacts such as foot morphology, skin and nail concerns, bony changes and/or pain/inflammation as well as injury data linked specifically to wearing high heels – these concepts important given the purpose of our investigation was to identify the relationship between choice of heel height as it relates to need/task. Therefore, we have altered the introduction in the first round based on Reviewer 1’s request, excepting inclusion of this one non-relevant article, instead introduced more relevant references in support of the suggested change. We have provided two additional sentences relating to this.

This includes page 4, line 70-71:

While there is no disputing that ankle equinus is damaging to foot health and function, there is no known direct link between the use of high heeled shoes and ankle equinus.

And at pages 4-5, Lines 77-85

However, a recent meta-analysis exploring high heeled shoe wearing and lower extremity biomechanics underlined the low levels of data available and the weak nature of measures of wearing impact. The cited review called for longitudinal research covering a wider range of heel heights to better understand their impact (18).

Reference:

Zeng Z, Liu Y, Hu X, Li P, Wang L. Effects of high-heeled shoes on lower extremity biomechanics and balance in females: a systematic review and meta-analysis. BMC Public Health. 2023;23(1):726.

2. The sample selection criteria (Barbie's screening criteria) are subjective. for example, the author's reference to limiting the number of Barbie dolls "for our own sanity" falls short of academic rigor. Please provide more objective screening criteria and remove subjective statements. For example, clearly explain why certain Barbie Dolls were excluded from the study.

Response. We have removed that statement. We provided a detailed screening criteria and explained how Barbie dolls were excluded as per the first round request, but have also added to these statements on page 6, Lines 145 to 154 as follows:

To ensure outcomes related to the international ‘mass market, we excluded any Barbie or friend that had limited international reach, such of those marked as special/limited/collector editions or aligning with Television series (e.g., Dolls of the World, Crayola, Sponge Bob Square Pants), those modelled on popular actual people (e.g., Audrey Hepburn), when Barbie was packaged together with Ken, Barbie’s younger sisters and their friends (e.g., Skipper), any who were Mermaids without legs/feet or ballerinas with fully moulded feet into shoes, Barbies with soft bodies or those with horses or vehicles. We also excluded Barbie released in only a single country or only one retail outlet (e.g., USA only releases for Sears, Target, or Walmart). We made these decisions to ensure we only categorised the most accessible, representative and mass produced Barbies available to people throughout the world. This ensured generalisability across all distribution points rather than Barbie dolls that were produced for a niche or single geographical market. This also ensured representation accuracy in referenced online catalogues and multiple reference points to support the validity of the developed data collection system. Many of the special/limited edition dolls, or those not widely distributed had limited online pictures which would have introduced potential misclassification of foot posture.

3. The statistical method part is not detailed enough, for example, the premise assumptions of correlation analysis (such as whether variables obey normal distribution) are not explained. It is suggested that in the statistical analysis part, the data processing process should be explained in detail, and the applicability and limitations of the analysis method should be discussed.

We have added additional detail. As per the first round, we provided, additional support of the limitation of the analysis methods and the reviewer has not provided comment on it’s acceptability, and what in the limitations is additionally required. We have also provided an outline of model choice where data was not normally distributed. This exploration of the two models is commonly encouraged when there are deviations to normality in limited variables and a large data set is encouraged allowing avoidance of overstating any observation, allowing robust analysis of limitations. We settled on Pearson’s to be the more conservative model.

We have added on pages 7-8, Lines 174-186:

There were no missing values. All data except for Foot posture (flat) and Equity (disability) were normally distributed. When exploring the kurtosis of the data, we found the Foot posture (flat) was slightly platykurtic and Equity (disability) to be leptokurtic. Spearman correlations were conducted to fully explore the impact of the limited data within the Equity (disability) and deviation to normality before deciding on a final model (28) Due to minimal differences between the Pearson and Spearman correlation coefficients and the Pearson correlations coefficients being more conservative, only the Pearson coefficients are presented (29).

Additional reference to support this approach:

29: Rovetta A. Raiders of the lost correlation: a guide on using Pearson and Spearman coefficients to detect hidden correlations in medical sciences. Cureus. 2020;12(11).

We added at the limitation in page 10, lines 312-316:

This research has its limitations. It is quite possible by limiting the Barbie collections to exclude special collections, the prevalence of Barbies with flat feet have been increased. Similarly, when comparing the correlation models, the use of Spearman correlation coefficient increased the strength of some relationships. Given the inclusion criteria, and minimal difference, we reported the more conservative of the two models.

4. Discussion section Some of the content is repeated, such as emphasizing the relationship between Barbie's foot position and career choice many times without exploring the possible mechanism, please simplify the discussion section, avoid repetitive content, and focus on the uniqueness of the study results and its scientific contribution.

Response:

We have removed some of the repeated statements both in the introduction and discussion. This can be seen in the tracked changes version.

We are, however, unsure what the reviewer means relating to possible mechanism? We have provided legislative change as a possible mechanism for this what is observed over time, and data on women’s workforce participation in the previous review round as a possible mechanism. However, if the reviewer is alluding to why high heel wear is prevalent, linked with fashion over occupation, this is still unclear. We have provided additional statements in support of this at page 10, page 301-304:

It is unclear why high heels are inherently linked with women fashion, given they originated in the male workforce and society (36). However, there are clear changes in wearing patterns over time and preferenced wearing of high heel footwear linked with psychosexual benefits (37).

Reviewer #3: This study exhibits serious flaws in methodological design and data reliability. The data sources primarily relied on publicly available catalogs, sales websites, and private collections, lacking scientific rigor and systematicity. This approach introduces significant selection bias and incomplete information risks, with no means to verify data authenticity or accuracy. Foot posture assessment was based solely on subjective image classification without objective measurements, further undermining the study’s credibility. The proposed FEET system, while innovative, lacks reliability and validity testing. Its classification criteria are overly simplistic, categorizing foot posture merely as "horseshoe foot" and "flat foot" without accounting for intermediate states or potential misclassifications. Data exclusion criteria were inadequately justified, particularly the arbitrary removal of special and limited-edition Barbie dolls, which lacks scientific rationale and severely limits the external validity of the findings. Statistical analysis relied exclusively on Pearson’s correlation coefficient, with no normality testing or control for confounding variables, rendering the methodology overly simplistic. In summary, the study suffers from critical weaknesses in data sourcing, research methodology, analytical approach, and ethical compliance, casting strong doubt on the reliability and scientific merit of its conclusions.

We have strong concerns about the genuineness of this review due to the factual errors in what has been written and will defer to the editor for response. In particular:

1. Publicly available data which is cross referenced, urls/linked provided in the manuscript, is transparent. This method of data collection allows this study to be replicated. This is a common method in systematic review and online research of publicly available data. As we provided all data sources and our raw data, this increases the robustness and authenticity of this research. We raise our concerns relating to this point as this reviewer ticked that data was not provided however we have provided our raw data as an appendix to enable its verification, authenticity and accuracy as per the requirements of PLOS One and good science. Similarly, the methods systematically describe how we obtained (with our sources), categorised, and then analysed the data.

2. The proposed FEET system – we acknowledge this is simplistic. However it is based on a doll and publicly available data which we tested within the team during development. We tested how repeatable our categorisations were (with an international and mixed gender team) and its validation.

3. The reviewer has used the term “Horseshoe foot” and as international researchers in foot health, we refute this is a foot type. We suspect this reviewer means Equinus – and this term potentially from a non-English translation or AI generated term. That said, Barbie only has two foot positions – one that can wear a high heel (equinus) and one that is flat. We used multiple data sources as referenced to classify these two foot positions. There are no additional foot types or measures as Review 3 requests, because Barbie is a doll and not a human.

4. We have provided normality statements within the methods and provided detailed limitations. Similarly, detailed why we have not included some dolls that are modelled on the female form.

5. We are unsure what issues the reviewer has with ethical compliance? We are using publicly available pictures of dolls that are modelled on the female form. There are no human or animal participants in this research. We researched dolls as a representation of humans, which means we cannot consent Barbie. However given the wider discourse on Barbie’s high heel wearing within the movie, her online marketing and footwear strategies, and we are confident she would have consented to this research.

---

## [Decision Letter · Decision Letter 2]

13 Apr 2025

Flat out Fabulous: How Barbie’s foot posture and occupations have changed over the decades and the lessons we can learn.

PONE-D-24-58543R2

Dear Dr. Williams,

We’re pleased to inform you that your manuscript has been judged scientifically suitable for publication and will be formally accepted for publication once it meets all outstanding technical requirements.

Kind regards,

Yaodong Gu

Academic Editor

PLOS ONE

Additional Editor Comments (optional):

Reviewers' comments:

Reviewer's Responses to Questions

**Comments to the Author**

1. If the authors have adequately addressed your comments raised in a previous round of review and you feel that this manuscript is now acceptable for publication, you may indicate that here to bypass the “Comments to the Author” section, enter your conflict of interest statement in the “Confidential to Editor” section, and submit your "Accept" recommendation.

Reviewer #1: All comments have been addressed

Reviewer #3: (No Response)

2. Is the manuscript technically sound, and do the data support the conclusions?

Reviewer #1: Yes

Reviewer #3: Yes

3. Has the statistical analysis been performed appropriately and rigorously? 

Reviewer #1: Yes

Reviewer #3: Yes

4. Have the authors made all data underlying the findings in their manuscript fully available?

Reviewer #1: Yes

Reviewer #3: Yes

5. Is the manuscript presented in an intelligible fashion and written in standard English?

Reviewer #1: Yes

Reviewer #3: Yes

6. Review Comments to the Author

Reviewer #1: (No Response)

Reviewer #3: This manuscript offers a novel and thought-provoking exploration of Barbie's foot posture from a cultural and biomechanical perspective. This interdisciplinary approach is creative and current, with clear relevance to discussions of gender, representation, and embodiment. This study provides valuable insights and opens up new avenues for future research.

7. PLOS authors have the option to publish the peer review history of their article (what does this mean? ). If published, this will include your full peer review and any attached files.

**Do you want your identity to be public for this peer review?** For information about this choice, including consent withdrawal, please see our Privacy Policy .

Reviewer #1: No

Reviewer #3: No

---

## [Editor Report · Acceptance letter]

PONE-D-24-58543R2

PLOS ONE

Dear Dr. Williams,

I'm pleased to inform you that your manuscript has been deemed suitable for publication in PLOS ONE. Congratulations! Your manuscript is now being handed over to our production team.

Kind regards,

on behalf of

Professor Yaodong Gu

Academic Editor

PLOS ONE